# Qualitative study investigating the professional and personal effects of patient suicide on general practitioners in Northern Ireland

Grainne McAnee ,[1] Kelly Norwood,[1] Gerard Leavey [2]

¹School of Psychology, Ulster University School of Life and Health sciences, Coleraine, UK
²Bamford Centre, Ulster University, Coleraine, UK

**Correspondence to**
Dr Grainne McAnee;
g.mcanee@ulster.ac.uk

## ABSTRACT

**Objective** There is a dearth in suicide literature addressing the impact on general practitioners (GPs) of losing a patient. We aimed to examine the personal and professional impact as well as the availability of support and why GPs did or did not use it.

**Design** A qualitative study using one-to-one interviews with participants recruited using snowball sampling.

**Setting** The study was conducted in a primary care setting.

**Participants** Interviews were held with 19 GPs within primary care in Northern Ireland.

**Results** GPs are impacted both personally and professionally when they lose a patient to suicide, but may not access formal help due to commonly held idealised notions of a 'good' GP who is regarded as having solid imperturbability. Fear of professional repercussions also plays a major role in deterring help-seeking.

**Conclusions** There is a need for a systemic culture shift within general practice which allows doctors to seek support when their physical or mental health require it. This may help prevent stress, burnout and early retirement.

## STRENGTHS AND LIMITATIONS OF THIS STUDY

⇒ The qualitative methodology employed in this study allows the depth of experience to be accessed in an area not widely researched in a population not comfortable discussing this topic.

⇒ The study explored experiences of patient suicide within the context of primary care in Northern Ireland, therefore these findings cannot be extrapolated to other medical settings and are not representative worldwide.

⇒ For a more inclusive picture, further general practitioners, psychiatrists, psychologists, nurses, community pharmacists and other professionals should be incorporated into the research to offer additional insight and understanding.

⇒ The emotional aspect of guilt and self-scrutiny following the death of a patient by suicide, may have contributed to potential under-reporting on the effects of patient suicide.

## INTRODUCTION

It is estimated up to 90% of people who die by suicide consult their general practitioner (GP) shortly before death, with 20%–76% seeing their GP in the month prior.[1] Impacts on GPs are both personal and professional.[2] Understanding these impacts may facilitate the implementation of support needed to allow GPs to understand and recover from patient suicide-induced trauma. There is a dearth in suicide literature addressing the reactions of GPs to this loss.[3–5]

A 2020 systematic review assessed the impact on mental health practitioners when they experienced this loss.[2] Guilt, blame, shock, anger, sadness and grief were common,[2] in addition to lowered mood, poor sleep and increased irritability.[6] Professional reactions included doubt about practice and decision-making,[7] more caution in the management of suicide risk[8] and consideration of retirement

or career change.[9] Similar research investigated the effects on frontline staff reporting that staff questioned themselves wondering if they could have helped more.[10]

Studies have examined the effects on psychiatrists and psychologists[11] with some focussing on GPs. In Rotar Pavlič *et al*'s qualitative study of Slovenian GP experiences (n=22) GPs needed emotional support after a patient suicide but reported barriers including lack of provision and personal hesitation due to perceptions of weakness and professional failure—leading GPs to only seek support from close colleagues.[4] Zambrano and Barton[12] reported that GP coping mechanisms included speaking with colleagues and acceptance of the death as unavoidable. A UK-based study using 198 semi-structured interviews with GPs noted a lack of institutional understanding and poor professional support.[5]

Another UK-based study focused on GPs' experiences of dealing with parents bereaved

by suicide while considering the impact on GPs.[13] They described personal grief at losing a patient for whom they cared. The stigma of GPs accessing formal support and reliance on colleague support were consistent themes. A survey of 152 rural GPs in the Ireland[14] who lost patients who died by suicide revealed GP reactions included feelings of guilt, sleep disruption, increased readiness to make psychiatric referrals in patient care and reduced confidence in decision-making.

In summary, GPs' experiences of suicide affect them on a personal level and inform how they care for future patients at risk of suicide. This study adds to the literature addressing how GPs react to the suicide of patients by exploring the impact at a personal and professional level using interviews with nineteen GPs in Northern Ireland. To the best of the author's knowledge, this is the first qualitative study of this scale in Northern Ireland. It is one which explored access to support in this population in depth. Finally, it allows the needs of GPs in Northern Ireland to be better responded to. It asked two main questions: (1) what was the personal and professional impact of patient suicide on GPs?; (2) what was the availability of support and the reasons for access or avoidance?

## METHOD
### Design
This qualitative study employed in-depth interviews (n=19) which were recorded and transcribed to gain an understanding of the impact of patient suicide on practising GPs, support that was available and support needed. Informed consent was given after comprehensive information on the topic was shared with potential interviewees. Experienced researchers were employed with interpersonal skills suited to the topic, should interviewees become distressed during interviews they were reminded they were allowed to stop the interview at any time should they find it too upsetting.

### Patient and public involvement
The project was supported by a patient and public involvement committee from its conception through to conclusion. The committee advised on all stages of the project. This included formulation of the research questions, design, recruitment and how the findings would be disseminated.

### Participants
The full study within which this is placed[15] interviewed both family members of people in Northern Ireland who died by suicide and GPs. GPs who were working in Northern Ireland and experienced the suicide of at least one patient in their care were invited to participate in the research by letter, with a follow-up phone call, according to the geographical location of their practice in order to represent any disparities between urban and rural areas. After a number of unsuccessful attempts to recruit GPs through traditional postal invitations, the 19 GPs who participated in the study were recruited through a snowball sampling approach with support from the Royal College of General Practitioners (Northern Ireland). The study explored[1]: contact, recognition and management of suicidal patients[2]; primary care liaison with psychiatric and other services; and[3] recommendations for improved care and suicide prevention

### Procedure and data analysis
Qualitative data from interviews were analysed using inductive, thematic analysis via NVivo12. Data were analysed via a six-phase process: data familiarisation, generating initial codes, searching for themes, reviewing the themes, defining and naming the themes and report writing.[16] Three researchers (KN, GMA, GL) read transcripts two times for familiarisation, then independently generated initial codes. Researchers independently coded transcripts in batches of five and then met to review and collapse codes. All transcripts were reanalysed using the final set of agreed codes from the iterative process. From these themes and subthemes were generated and reviewed until consensus was reached.

### Reflexivity
Qualitative researchers are required to make sense of the material they engage with[17] and as such it is imperative that they acknowledge how their own experiences and beliefs can impact their perspective. With such an emotive and delicate topic it was important to be mindful and take appropriate steps against this. Participants provided data on experiences which are powerful and important as well as deeply personal and the respect it was treated with included that it was approached with sensitivity and objectivity. It was important that researchers remain open to the possibility of findings being contradictory to personal feelings and beliefs.[18] Qualitative research can involve the practice of analysts disregarding prior knowledge and experience;[19] however, this may not always be possible or indeed necessary.[17] The researchers adopted a practice of self-reflection and consensus of three researchers to protect robustness of findings. The use of a literature review before analysis and placing findings within the context of that literature also protected the integrity of analysis.

### Trustworthiness and credibility
Due to lack of standardisation of the methodologies employed in qualitative research, it is not possible to conduct formal reliability testing however Savin-Baden and Major[20] provide criteria to improve trustworthiness which include methodological coherence (congruence between research objectives, methodologies, data and analysis), audit trail (description of the research process from beginning to end which can uncover other possible influences on findings) and negative case analysis (actively identifying and extracting data which was not compatible with the main or expected findings in order to encourage critical thinking and subsequent modification). The use

**Table 1** Summary of themes and subthemes generated from general practitioner interviews

| Themes | Subthemes |
|---|---|
| Personal reaction | Viewing a body<br>Short-term feelings<br>Long-term feelings |
| Professional impact | Analysis of events<br>Improvements to practice<br>Blame culture |
| Barriers to accessing support | Knowledge of services<br>The hidden contract<br>Professional repercussions |
| Psychological support | Family and friends<br>Within the practice<br>Other services |

of the Standards for Reporting Qualitative Research was adopted to further the aim of trustworthiness.[21]

## RESULTS

Nineteen GPs took part (n=11 women and n=8 men) from a range of primary care settings, (socioeconomic and urban, semi-urban and rural populations). Participants were long-serving practitioners (15–30 years' experience), involved in the care of more than two patients who died by suicide, several GPs indicated more than 10 cases, and three had personal experiences of suicide.

Analysis of the data generated four themes; personal reaction, professional impact, barriers to accessing support and psychological support. Themes and subthemes are shown in table 1, figure 1 shows the relationships between them.

Most GPs had been in practice for more than 20 years with knowledge of multiple suicides, in practice and in their personal lives. GPs had experienced the loss of siblings, other family members, friends and colleagues. They supported family members of patients, close colleagues and other members of their practice. The impact of suicide was variously described as 'devastating' or 'very painful'. No GP was unaffected by the suicide of a patient and effects were negative. Several GPs became distressed during interviews.

Five patient suicide typologies were interwoven into the experiences recounted[1]: avoidable deaths: regarded by GPs as people who might still be alive had services acted more swiftly or were better coordinated[2]; missed signs: suicides signalled by the patient but the GP recognised only in hindsight.[3] Valued patients: suicide of patients who had coped with chronic illness with cheerful stoicism[4]; apparent improvement: suicides that occur despite apparent good mood and social functioning[5]; family complaint: GP feels scapegoated by the family.

### Personal reaction

Most GPs reported that the suicide of a patient 'has a massive impact on the clinician as a person' (GP11). The work of the average GP means that they will encounter what one described as 'endless human misery', with a GP describing the impact:

> Your whole life is absorbing. … you absorb everything … everybody's pain and sorrow … (GP9)

Feelings expressed by GPs when they lost a patient included shock, loss, anger, upset, distress, desperation and sadness. The strongest feelings were responsibility, guilt and failure, 'there is an awful lot of guilt and everybody feels guilty' (GP12). Several GPs said they had experienced periods of depression and had witnessed colleagues being 'destroyed' in the aftermath.

GPs expressed that they found the process of confirming the death to be 'very, very troubling' (GP2) and in a manner resonating with post-traumatic stress disorder, described how it became 'a very visual memory' (GP2) which remained with them. After this initial reaction came short-term feelings. One GP talked about the suicide of an elderly patient who had started exhibiting

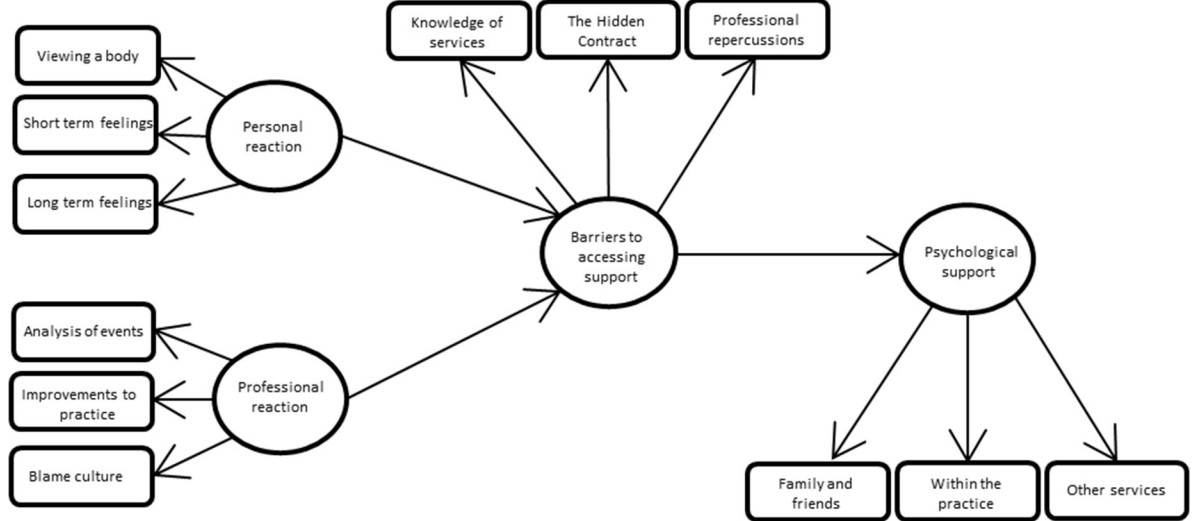

**Figure 1** Thematic map showing relationship between themes.

erratic behaviour. The patient was engaged with services at the time with professionals disagreeing over a diagnosis. The GP felt the patient had fallen into 'the gaps of the system' (GP11). The short-term impact on the GP was devastating.

> I came in and did my work and I went home and I cried, probably for about two weeks, all night. (GP11)

Short-terms feelings were led by a period of intense reflection after the loss during which they ruminated on the events preceding the suicide because 'there is always a huge guilt issue around suicide' (GP7). One GP sums up this dynamic and the need for reassurance.

> You want to know it's not your fault. You feel guilty because there's always something you could do … or maybe there isn't but you always feel that. (GP4)

Different conclusions were reached seeking to quiet that voice that suggested that 'there's always something you can do' (GP1). These depended on assessment of prior events and relationships between the patient, GP and other services. Where there had been a history of self-harm, alcohol or substance abuse, poor engagement with services or impulsive behaviour, GPs indicated that these deaths were upsetting but not shocking and they could not have changed the outcome; 'when you reflect on what could you have done differently that might have had an impact, most of them you couldn't' (GP5). This recognition did not always permit feelings of failure to be resolved.

Other GPs concluded the outcome could have been changed. One GP working in an area of high deprivation reflected on the death of a young patient with a history of psychiatric problems who had been seen by several services in the days leading to their suicide. The patient stated they were suicidal and had shown signs of being suicidal during a home visit. Although services were alerted, the patient died by suicide. The GP felt that the system had failed.

> I was very upset about it because I felt that if he had been immediately sectioned he would have been alive … (GP1)

Others had the anxiety of never knowing if alternative action could have altered the outcome. One GP talks about patients who 'leave no footprints in the snow' (GP19). This GP is describing patients who show no signs of considering suicide and leads the GP to wonder both,

> is it a switch that can flick in your head? And the natural follow-up from that is could that flick in anybody's head … (GP19)

The description used is suggestive of something very haunting with a sense of silence wrapped around it. Another GP believed that,

> … most people who end up committing suicide, there probably is something documented … about their mood. (GP14)

One GP described the experience of losing a young person who had previously presented with suicidal ideation. Although in treatment they showed no signs of being actively suicidal; they regularly attended the practice but consultations were about minor physical issues. The loss of the patient 'was not a predicted suicide' (GP12) and it 'really shocked the whole practice' (GP12). Hindsight revealed signs that were not recognised and they were left feeling that 'in retrospect maybe things were more difficult' (GP12) than the patient suggested. These questions remained unresolved, leaving feelings of guilt.

One difficulty expressed by GPs was the non-return of a patient whose mental state was concerning. GPs may assume that the problem is resolved, for patients the opposite may be true.

> I suppose as a GP you always assume that patients don't come back to see you if they're better, and that often is not the case, especially with mental health. (GP3)

Part of the reason that GPs do not follow-up is due to the nature and management of their workload. GPs describe an overloaded day with little room to spend time with patients and none for activities like following up patients they may have had a concern about; a 'niggle'. GPs do not know if they could have prevented a suicide but they also don't know the optimal criteria for intervention, how or when to intrude—'you're never going to find out' (GP4). It can be very hard to know when the steps taken saved a life, but they will always know when they did not.

Long-term feelings sometimes meant that even years after, the suicide was experienced as a painful memory. One GP described the 'intensely harrowing' experiences of a close colleague who had experienced several 'traumatic suicides', as a 'thinly covered scar' (GP6) implying a wound that has never really healed. Another long-time impact is when other suicides occur 'it brings back the memories then of the others' (GP7).

## Professional impact

There was a professional impact of a patient suicide. For most these effects provided the catalyst to change the practice management of suicidality, but several described how patients' suicide had provoked early retirement of colleagues. We noted considerable differences in how the aftermath was managed. Some practices had standard procedures such as conducting an event analysis to see 'whether we could have done anything differently or we could change our system' (GP14) or meeting with other professionals involved for reviews. Even if there was no formal procedure, GPs performed some level of analysis. A GP who had experienced the death of their elderly patient described how they reviewed their notes with 'a

fine-tooth comb' (GP11). These processes allowed GPs to explore and put in context the events leading up to the suicide.

There are complicated personal and social factors that influence GPs judgement and decision-making. In relation to risk management, one such factor was knowing the full picture. Some GPs became aware patients were telling families a different version of events than in consultations. One GP described how a family member was given 'a very different picture' (GP12) compared with the GP around the patient's state of mind and what help had been both offered and rejected by the patient. Another GP revealed that they may have 'a lot of people pretending to be mentally ill to get disability living allowance' (GP9).

### Commonly reported impact was heightened vigilance

> I would say I am much more vigilant and I'm now not the little kind of shy GP who is afraid to ask questions. (GP7)

This extended to risk factors outside of the patient themselves such as a previous suicide in the family. It was not restricted to patients who presented with mental health issues, for example, it could be extended to patients in the same age range as the patient who had died.

> if I have a younger person in consultation now, I am looking at the mental health issues … prior history of any mental health issues (GP12)

Commonly expressed and contributing to feelings of guilt and responsibility was 'blame culture'—GPs felt they were being held accountable for patients' ill-health and their 'failure' to cure people often resulted in blame. GPs found this demoralising and professionally devaluing.

> People with no medical education, people with too much medical education, but everyone loves bashing GPs (GP4)

One GP who treated a young man shared the experience of being blamed when he died. The parents openly censured the GP, reproaching them for not doing more. The GP felt the parents had unrealistic expectations of primary care services and the extent of help available:

> … there was huge blame. The parents felt that we should have gone to their house and done more. (GP6)

### Barriers to accessing support

Given the impact described above it was worrying that GPs reported concerns and described barriers to accessing formal support. Three subthemes were generated. First, differing knowledge between GPs of available services. Some believed support by occupational health was available, others were sure it was not. One reason that GPs did not fully understand what help is available to them

professionally is because they would be highly unlikely to consider using it. Even when there was knowledge about support services available, GPs stated it would not be accessed.

> The BMA have a good counselling service by telephone … it never occurred to me to use it. (GP6)

A second subtheme was that for GPs it is not acceptable to admit that sometimes those who help others, need to be helped themselves.

> GPs don't go sick. They work through ridiculous things because there's this macho "we're tough". (GP6)

An over-valued work ethic or sense of duty was referred to as 'the hidden contract' or 'the hidden curriculum'—described as an understanding that 'I will not tell you if I feel bad and I will not ask you if you feel bad'. It is a cultural concept that is absorbed during training. It appears to be strongly associated to an internalised perception of the GP as a healer, a source of strength, but not someone who may need support and care. While at one level this appears to be a deeply held cultural trope, on a more direct level, GP mental health issues may impinge on the doctor's fitness to practice.

In the final sub-theme we noted concerns related to professional repercussions if a GP accessed help related to their own mental health and/or ability to cope. They worry that their fitness to practice (and their career) will be jeopardised.

> … if you had a sore knee, that wouldn't be an issue; but if you admit to depression there is a big question mark about your fitness to practise. (GP10)

The GPs overwhelmingly indicated they would not consider using support but they also said they knew it was needed;

> we need more support when suicides have happened … it needs to be built into the culture that we accept things when these things happen. (GP6)

### Psychological support

Given the intensity of GP's responses to losing a patient to suicide and the barriers reported above, the next question becomes centred on what support was available in assisting doctors to process the experiences and reduce negative impacts on future practice or potential premature retirement. If a GP does decide they would like support, instead of a robust and easily accessible GP support system it appears that the health system makes scant allowance for the impact of a traumatic event on the GP; most participants suggested that they are obliged to just get on with it.

… but pull up your socks and get on with it. This is your job; get on with it. You have to deal with this. (GP11)

There was a strong message that it is hard for GPs to seek professional help for this loss and a sense of conflict around it.

I'm not saying GPs don't need it, I'm saying that GPs don't access it, and the reason I think they don't access it is twofold. I suppose GPs feel that I'm big enough … I'm strong enough, I don't need counselling. And second of all, there's an embarrassment issue of having to seek help yourself as someone who gives help to others. I think there is a difficulty there. (GP1)

GPs stated a lack of concern or compassion for them in the immediate aftermath, one describing an expectation that they would be able to continue working with no reaction to what they describe as a traumatic event,

there's a huge difference in terms of hearing that someone has taken their own life and then being the GP who's actually going out to confirm that person has passed away, and I think it's one of the things, that it is fairly traumatic to see the person in that situation. (GP2)

They found this expectation uncaring and resented it,

The thing is we came back and just worked the rest of the shift … I just remember coming in the next day and no-one even saying, "that must have been tough". (GP2)

The most often used support was informal. Some GPs found support within their own homes, for example married GPs found support from their domestic partner, but for most GPs they found support within their own practice.

I suppose in my practice we have three GPs. We are all much of the same age group and we would be supportive of each other, and certainly if we have a difficult case we would talk about it very freely with my two colleagues, and I think that acts as a sort of a debriefing type of intervention in itself. (GP1)

… it has a big impact on how you deal with other patients in your day to day practice. Your concentration is not the same and you relive things and increased risk of mistakes and lack of care and lack of concentration so you're not really giving your patients 100%. But I didn't seek help then, but my partners were brilliant, they were very supportive and they were very good and we talked about it a lot. (GP11)

GPs described how general practice can be 'reasonably isolated'. One reported that they felt the impact of a patient suicide was more damaging for them than for other mental health professionals such as psychiatry. The GP is seen as 'all things to all men' (GP6), a person who

can cure anything. There is the expectation in psychiatry that no matter how hard you try, you will lose patients in this way 'whereas in general practice we don't find it easy to accept failure' (GP6). GPs are closer to the local community, have personal contacts and may be involved with care of other family members.

you're not just involved with that person but you're involved with the whole family, and that's real general practice … (GP7)

Perhaps the strongest message from this theme is that no GP reported that they used professional support.

To conclude, GPs undergo complicated experiences when they lose patients to suicide. They spend time in intense personal and professional reflection. They may feel that a patient was lost who did not need to be or conclude that they could not have helped. GPs are left with a profound and lasting sense of guilt, responsibility and failure. They may blame themselves for the death, they may be blamed by others. Mistakes may have been made. This is often channelled into taking action to try and ensure it does not happen again. Sometimes it does and the need for support increases. GPs know that they need support, but they don't feel they can access it. The age-old question remains, 'who heals the healer?'.

## DISCUSSION

Previous studies have examined the impact of patient suicide on psychiatric staff but fewer studies have examined specifically GPs reactions. And none to the author's knowledge in Northern Ireland to the depth that this study did. We sought to explore personal and professional impacts, and support systems available for GPs who experienced patient suicide. We generated four themes: personal reaction, professional impact, barriers to accessing support and psychological support. The themes are similar to those uncovered in studies of suicide impact on other medical professionals. In terms of personal reaction, experiencing a period of rumination or 'great self-scrutiny' as described by Saini *et al*[5] with similar personal and professional impacts. Guilt, blame, shock, anger, sadness and grief following the suicide of a patient are commonly reported.[2] GP distress was noted by Foggin *et al*[13] and Saini *et al*.[5] The current study additionally indicates the considerable psychological and organisational barriers to GP support following patient suicide. We noted a significant dissonance between the GP self-image of solid imperturbability, and the distress caused by patient suicide. The idealised estimation of a 'good' GP embodied in a 'hidden contract' develops in training and is reinforced over years but in practice but appears to fail in the long-term as GPs gain experience and insight into when they need to seek help and form mechanisms to cope with difficult days. A level of objectivity or detachment from the patient is a form of self-protection that may appear sensible and attractive in training but deteriorates over time. One GP contrasted the loss of a patient to

suicide in general practice to that experienced in psychiatry where there is a commonly held expectation that some patients will die by suicide. Regular contact with high-risk patients increases this expectation. In primary care where suicide is relatively rare a patient's suicide is more likely to undermine the professional identity of GPs who work as individuals rather than part of a clinical team.

The 'hidden contract' discourages GP help-seeking for psychological support. It is seen as a weakness. This fear of being diminished by loss of regard by others has been described in other studies of self-care and intra-professional support among GPs and other high-visibility caring professions in which coping, self-reliance and resilience may be over-valued to the detriment of the individual.[22 23]

GPs describe a high-pressure environment with an excessive workload all swiftly managed in consecutive ten-minute appointments. In this context, it is difficult to provide compassionate care which requires time to listen in order to understand. GPs are conflicted by systemic requirements for example they might wish to request hospital admission for people whose mental health has deteriorated knowing this is in opposition to the policy of community care. Other role conflicts are the amount of time spent with individuals should be dictated by patients' needs but cannot be due to time demands and pressures. GPs may want to acknowledge the distress of bereaved patients but seek to convey a detached professionalism.

Cheshire et al[24] used a qualitative design to understand how GPs coped with workplace challenges and stresses. Among these were the inability to connect authentically due to time demands and requirements such as the need to record patient indicators within 10 min appointment. GPs felt primary care was functioning in more detached ways which are more comparable with secondary care functioning. Some GPs felt that working longer hours was the most effective way they could deal with the sense of being overwhelmed, to the detriment of a work-life balance. Strategies which the GPs used to mitigate work stress included meditation, mindfulness, stress management techniques, taking regular exercise, eating well and meeting and debriefing with their colleagues, however the study also reported that a surprisingly high number of participants had adopted changes such as reducing their working hours or changing their role to locum to allow them to downsize and regain work-life balance.[24]

Emotional hardening and cynicism among GPs appeared ubiquitous. An example of why this may be was provided by one GP who described going to identify a body after a suicide and on their return to a busy day and a list of patients, 'no-one even asked' how they were, suggesting a resigned fatalism for at least one GP in this sample. Perhaps exacerbating the self-neglect reported was that GPs were concerned about the professional repercussions that might accompany accessing support. GPs raised the issue of potentially being declared unfit to practise if formal help was sought, a finding supported by

previous research. Spiers et al[25] found that GPs reported issues around privacy, confidentiality and the stigma surrounding seeking help. This worry around professional repercussions begins with medical training and is reported by medical students.[26]

The lack of uptake of services for mental health issues in the wake of the loss resonates with low GP uptake of care when physically ill in a study reported by Thompson and colleagues.[23] GPs want help but do not feel they can access it. The impact of this inability to seek help is seen in studies conducted on the mental health of doctors. Doctors experience high rates of anxiety,[27] depression,[28] substance misuse,[29] emotional exhaustion and burnout[30] and are at high risk of suicide themselves.[26] Male doctors are 1–4 times more likely than other men to die by suicide and female doctors 2–3 times more likely than other women.[31] This study showed that the impact of loses of this nature may even lead a GP to consider leaving the profession which has implications on personal, practice and regional service provision levels.

GP uncertainty about the availability of support may indicate their own ambivalence about help-seeking or the inappropriateness of current available support. While GPs are seen as part of the National Health Service in the UK, they are individual practitioners who work as self-employed practitioners within GP surgeries. One review of the literature recommended that doctors require specialist services which focus on the unique and complex situation of a doctor becoming a patient.[32] George et al[33] outlined a pilot aimed at preventing or minimising mental health issues in doctors. This strategy would operate at a primary, secondary and tertiary level. Primary level prevention would be aimed mainly at medical students and may work to counteract the macho culture in which a doctor is expected to show no weakness. This level would include awareness-raising, enhancing coping skills and stress management skills. Secondary level prevention would allow early diagnosis and intervention and would include measures for screening, reducing stigma and creating a supportive culture for doctors. The tertiary level would include offering treatments and interventions to minimise the chronic negative effects of mental ill health on doctors.

Strengths of this study lie in the qualitative methodology employed in an area where depth of experience is complicated, nuanced and important to understand. The study allowed GPs to talk in depth about a subject matter which is both stigmatised and sombre. This allowed responses to be accessed in an area not widely researched in a population not comfortable discussing this topic. Findings showed that GPs are not comfortable but nevertheless need to talk about this loss, and this study provided a forum for them to do that, one which would not normally be presented. Limitations include that findings cannot necessarily be extrapolated to other medical settings outside of primary care and are not representative worldwide as care systems vary and different systems will offer different responses. However, there is bound

to be something fundamentally similar in the reaction to this loss from someone who has dedicated their life to helping others. To know rather than surmise, further GPs, psychiatrists, psychologists and other suicide professionals should be incorporated in future research to offer additional insight and understanding. It is important to note also that the emotional aspect of guilt and self-scrutiny following the death of a patient by suicide may have contributed to potential under-reporting on the effects of patient suicide. Only by making the changes needed which will reduce the stigma around this issue can we know the scale as well as the nature of the impacts.

## CONCLUSIONS

The findings of this study suggest a need to provide GPs with the appropriate psychological support after losing a patient to suicide. Despite a wish to remain true to the GP self-image of solid imperturbability created by adherence to the 'hidden contract', GPs report that the loss of a patient to suicide results in a period of self-scrutiny. From this may come short or longer term effects on their mental health. The 'hidden contract' translates into doctors being unable to comfortably make a transition to patient which means that doctors are unable to properly receive the care they seek to provide for others.

Creating a culture shift which allows doctors to complete this transition will help prevent burnout and early retirement from a system which is not currently supporting them. One way to help create that change is to include an opposing view as part of training for all health and social care professionals for example by including in the curriculum modules on how to understand and deal with the effects of self-harm and suicide. The price for seeking that help should not include that their fitness to practice will be questioned consequently. GPs work in a high-risk, high-stress environment in which they are likely to experience loss which will require intervention and support. We recommend the provision of a strategy which will facilitate the culture change needed and implement the services needed to support our doctors in general practice, including guidelines for GPs on when and how to ask for help. If a change is not facilitated, we will continue to see GPs experiencing distress and mental health issues which when unaddressed are contributing to mental illness, burnout and early retirement.

**Contributors** GL acted as guarantor for the study. GMA, KN and GL were responsible for data analysis and interpretations. GMA wrote the initial draft of the article. GMA revised the article in response to the comments of the coauthors and all subsequent reviews on all drafts. All authors gave their final approval of the version to be published.

**Funding** This study was funded by the Northern Ireland Public Health Agency (Grant COM/4032/08). The funder played no part in the conduct or reporting of the study, nor had any involvement in the interpretation of the data.

**Map disclaimer** The inclusion of any map (including the depiction of any boundaries therein), or of any geographic or locational reference, does not imply the expression of any opinion whatsoever on the part of BMJ concerning the legal status of any country, territory, jurisdiction or area or of its authorities. Any such expression remains solely that of the relevant source and is not endorsed by BMJ. Maps are provided without any warranty of any kind, either express or implied.

**Competing interests** None declared.

**Patient and public involvement** Patients and/or the public were involved in the design, or conduct, or reporting, or dissemination plans of this research. Refer to the Methods section for further details.

**Patient consent for publication** Not applicable.

**Ethics approval** This study involves human participants and the study 'Understanding Suicide and Help-Seeking in Urban and Rural Areas in Northern Ireland' received ethical approval from the Office of Research Ethics Committees Northern Ireland (REC reference: 10/NIR03/65) and the main report provides full details of the methodology. Participants gave informed consent to participate in the study before taking part.

**Provenance and peer review** Not commissioned; externally peer reviewed.

**Data availability statement** No data are available.

**ORCID iDs**
Grainne McAnee http://orcid.org/0000-0001-7661-1607
Gerard Leavey http://orcid.org/0000-0001-8411-8919

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
