## [Reviewer comments · BMJ Open]

ARTICLE DETAILS

TITLE (PROVISIONAL)	A qualitative study investigating the professional and personal effects of patient suicide on General Practitioners in Northern Ireland.
AUTHORS	McAnee, Grainne; Norwood, Kelly; Leavey, Gerard

VERSION 1 – REVIEW

REVIEWER	Sahm, Laura University College Cork, Pharmacy
REVIEW RETURNED	14-Sep-2023

GENERAL COMMENTS	Thank you for the invitation to review this paper; General Practitioners and patient suicide: A qualitative study investigating the professional and personal effects. This is a very well conceptualised and written paper. I found it to be very interesting and important work, even though it is a sombre subject. I have a very few minor comments Abstract: Participants Interviews were held with 19 GPs who have experienced the loss of two or more patients to suicide. This I believe is part of methods and so should rather be confined to the inclusion criteria rather than giving results. Page 5: For a more inclusive picture, further GPs, psychiatrists, psychologists and other suicide professionals should be incorporated in the research to offer additional insight and understanding. My suggestion would be whether this could be made even more inclusive of healthcare (rather than solely suicide) professionals, e.g. nurses, community pharmacists Page 7: They described personal grief at losing a patient they had cared for. Consider, at losing a patient for whom they cared. Page 8: Nineteen GPs took part (n=11 women and n=8 men) from a range of primary care settings, (socio-economic and urban, semi-urban and rural populations) recruited via snowball sampling. Participants were long-serving practitioners (15-30 years' experience), involved in the care of more than 2 patients who died by suicide, several GPs indicated more than 10 cases, and three had personal experiences of suicide.
---

	In my opinion these are results and so I would describe the participant inclusion / exclusion criteria and move the results to the appropriate section. Page 19: Emotional hardening and cynicism among GP appears ubiquitous. Would make this past tense “appeared” Page 19: Perhaps exacerbating the self-neglect GPs were concerned about the professional repercussions that might accompany accessing support. Please rephrase as I find this could be clearer. Finally I have a recommendation for the conclusion, which the authors may choose to include as they see fit. I am currently involved in an undergraduate curriculum for health and social care professionals in which we are seeking to bring this sombre subject of suicide and self-harm, and potentially skills to cope with it as a new module. This may be a suggestion as to how to stop future GPs and HCPs from feeling that they have to cope all alone.
--	---

REVIEWER	Mughal, Faraz Keele University
REVIEW RETURNED	19-Sep-2023

GENERAL COMMENTS	Thank you for the opportunity to review this interesting manuscript. I have made several suggestions which I hope the authors will attempt to incorporate to the improve reporting of their study. I think this study is important but it needs some work before publication. Introduction: In the first sentence 'complete suicide' is used. Please amend this to more appropriate language such as 'died by suicide'. They can refer to Samaritans or IASP guidance on language used when writing about self-harm and suicide. Can the authors describe some of the impacts of suicide in a patient on GPs. I note in the third paragraph authors write that 'GPs needed support' - can they be more specific in what support or needs were identified. What was the survey of GPs in Ireland about? It is not clear - was it after a death by suicide in practice? Note 'GPs' experiences'. It doesn't quite follow that experiences of GPs after suicide may influence them to leave the profession. I would either remove this entirely, or provide a strong rationale for why this may be the case with references or move to discussion where I now can see why this occur from the data. So, is the niche of this study the fact that it is the first study in NI? If so, state this. Was it that access to support was explored in depth? Does it enable NI to understand how better to respond. We don't really need another study as the authors have described quite a bit of work already in this area...point is - clearly state novelty of this study. Methods Page 7, line 52 - suggest reword of 'this study employed qualitative data analysis using in-depth interviews' to 'this study employed in-depth interviews to....'
--

	There is a lot in the participants subsection that belongs in the findings - the sample characteristics. Line 20 page 8 about the wider study is a bit of place. Needs more context. Are lines 27-29 needed as the authors have already said this in the paragraph above. More detail needed on sampling method - how was recruitment achieved. Expressions of interest. Timescale etc. 'Repetition of text within transcripts facilitating....' does not make sense to me. Can this be reworded please. Are you meaning common or recurring codes? Themes are also 'generated'. This is now B&C's preferred term over 'developing' or 'emerging', and it is technically more accurate as it intends an active process of delving into the data iteratively and collaboratively to generate themes. The talk of main themes, broader themes, and sub-themes is confusing. I suggest the authors reword this. Perhaps they mean higher categories were developed before initial candidate themes, and then final themes after review and refining - see B&C on reflexive thematic analysis 2019 or 2022 and what they suggest on describing theme generation. Results Themes are not 'identified' from the data, but 'generated' - see latest B&C work. I note GPs became distressed in interviews - was there a study risk protocol - how did the study team handle this. This needs describing for ethical purposes. Patients who leave no footprints in the snow is very powerful and I can resonate with this in NHS general practice - perhaps expand a little here. 'Another GP believes that' - should it be past tense? First line under personal reaction - was this some or most GPs - worth adding to let reader know a few or a lot Under professional impact 'patients' suicide' is grammatically incorrect. The following sentence does not make sense - line 11-12. Can the authors elaborate on the complicated personal and social factors that influence GPs judgement and decision-making. The data under psychological support is interesting but feels a superficial to me. Are there are more data extracts the authors can incorporate to tell a richer story under this theme. Some of the quotations are also short - some of them would benefit from more data in them. Page 16 line 3 be clear you are talking about a subtheme here. The final theme is clear and coherent and well written. I like the summary. Discussion I wonder if rumination and personal impact could be combined. The first three themes feel a little superficial and there are interconnections. It may need more analysis or more expansive analytical narrative. A thematic map would help also. 'Unperturbability' - should it not be imperturbability. I think framing the hidden contract as failing with experience and more contact with patients is incorrect. I think it is more that GPs with experience gain more insight into when they need to seek help and form personal support circles and mechanisms to cope with difficult patients and hard days. GPs still practice with a sense of objectivity because they need to assess and manage patients accurately.
--	--

	I would not call one GP account as a resigned fatalism. It could be a deviant extract or anomaly. Would need to be a theme occurring over several samples of GPs to state something that strong about the profession as a whole I don't see mention of the strengths and limitations of this study outlined in the discussion There are also no future research recommendations mentioned or suggested. Experiences of accessing support would be good as well as GP specific brief interventions for GPs in distress but find where the gaps are. Conclusion This is written well and I don't have anything to add. I agree with the authors thoughts here. A culture change is needed. In England there is a NHS practitioner health service which I have heard has had a lot of GP self-referrals and allows doctors to seek confidential private help from NHS clinicians in group or one to one format. GPs unfortunately are under immense stress and strain in a wilting primary care system in the NHS. Workload is rising and resources are reducing. At present, without sustained acute investment, it is not looking bright for the general practice workforce. Unfortunately there have even been a few recent GP suicides which is a tragedy, just like anyone who dies by suicide. I enjoyed reading this manuscript. The methods need reworking and clarifying and I think some of themes can be further matured and developed. A thematic map would help with interconnections shown. Overall, well done to the 1st author, and the team. I hope these comments are useful. This study should add weight to the growing literature of the impact of suicides on GPs.
--	--

VERSION 1 – AUTHOR RESPONSE

Reviewer 1	
Abstract	
Participants	
Interviews were held with 19 GPs who have experienced the loss of two or more patients to suicide. This I believe is part of methods and so should rather be confined to the inclusion criteria rather than giving results.	Updated as requested.
Page 5: For a more inclusive picture, further GPs, psychiatrists, psychologists and other suicide professionals should be incorporated in the research to offer additional insight and understanding. My suggestion would be whether this could be made even more inclusive of healthcare (rather than solely suicide) professionals, e.g. nurses, community pharmacists	Updated as requested.

Page 7: They described personal grief at losing a patient they had cared for. Consider, at losing a patient for whom they cared.	Updated as requested.
Page 8: Nineteen GPs took part (n=11 women and n=8 men) from a range of primary care settings, (socio-economic and urban, semi-urban and rural populations) recruited via snowball sampling. Participants were long-serving practitioners (15-30 years' experience), involved in the care of more than 2 patients who died by suicide, several GPs indicated more than 10 cases, and three had personal experiences of suicide. In my opinion these are results and so I would describe the participant inclusion / exclusion criteria and move the results to the appropriate section.	Updated as requested.
Page 19: Emotional hardening and cynicism among GP appears ubiquitous. Would make this past tense "appeared"	Updated as requested.
Page 19: Perhaps exacerbating the self-neglect GPs were concerned about the professional repercussions that might accompany accessing support. Please rephrase as I find this could be clearer.	Yes, this does not make sense. Reworded. Thank you.
Finally I have a recommendation for the conclusion, which the authors may choose to include as they see fit. I am currently involved in an undergraduate curriculum for health and social care professionals in which we are seeking to bring this sombre subject of suicide and self-harm, and potentially skills to cope with it as a new module. This may be a suggestion as to how to stop future GPs and HCPs from feeling that they have to cope all alone.	Excellent. Definitely wish to include this.
Reviewer 2	
Introduction:	
In the first sentence 'complete suicide' is used. Please amend this to more appropriate language such as 'died by suicide'. They can refer to Samaritans or IASP guidance on language used when writing about self-harm and suicide.	Thank you for this. Updated as suggested.
Can the authors describe some of the impacts of suicide in a patient on GPs. I note in the third paragraph authors write that 'GPs needed support' - can they be more specific in what support or needs were identified.	Clarified that they required emotional support.

What was the survey of GPs in Ireland about? It is not clear - was it after a death by suicide in practice?	It was. This was clarified.
Note 'GPs' experiences'. It doesn't quite follow that experiences of GPs after suicide may influence them to leave the profession. I would either remove this entirely, or provide a strong rationale for why this may be the case with references or move to discussion where I now can see why this occur from the data.	Fair comment and yes it did arise from the data so have restricted this to the discussion.
So, is the niche of this study the fact that it is the first study in NI? If so, state this. Was it that access to support was explored in depth? Does it enable NI to understand how better to respond. We don't really need another study as the authors have described quite a bit of work already in this area...point is - clearly state novelty of this study.	All of these! Added to the introduction section.
Methods Page 7, line 52 - suggest reword of 'this study employed qualitative data analysis using in-depth interviews' to 'this study employed in-depth interviews to...'	Updated accordingly.
There is a lot in the participants subsection that belongs in the findings - the sample characteristics.	Updated accordingly.
Line 20 page 8 about the wider study is a bit of place. Needs more context.	Agreed. Updated accordingly.
Are lines 27-29 needed as the authors have already said this in the paragraph above.	Fair. Removed.
More detail needed on sampling method - how was recruitment achieved. Expressions of interest. Timescale etc.	Participant section updated.
'Repetition of text within transcripts facilitating....' does not make sense to me. Can this be reworded please. Are you meaning common or recurring codes? Themes are also 'generated'. This is now B&C's preferred term over 'developing' or 'emerging', and it is technically more accurate as it intends an active process of delving into the data iteratively and collaboratively to generate themes.	Have reworded to 'Main themes were dictated by repetition of similar experiences within the transcripts.'
The talk of main themes, broader themes, and sub-themes is confusing. I suggest the authors reword this. Perhaps they mean higher categories were developed before initial candidate themes, and then final themes after review and refining - see B&C on reflexive thematic analysis 2019 or 2022 and what they suggest on describing theme generation.	I agree, this needs tidied up. Used Byrne, D. (2022). A worked example of Braun and Clarke's approach to reflexive thematic analysis. Quality & quantity, 56(3), 1391-1412 to clean up the description.
Results - Themes are not 'identified' from the data, but 'generated' - see latest B&C work.	Updated accordingly.

I note GPs became distressed in interviews - was there a study risk protocol - how did the study team handle this. This needs describing for ethical purposes.	detail added to the Design section - page - to outline how a GP becoming upset was responded to.
Patients who leave no footprints in the snow is very powerful and I can resonate with this in NHS general practice - perhaps expand a little here.	Delighted to, trying to balance the richness of this very compelling data against the word limit.
'Another GP believes that' - should it be past tense?	Updated accordingly.
First line under personal reaction - was this some or most GPs - worth adding to let reader know a few or a lot	Removed overwhelmingly and replaced with most.
Under professional impact 'patients' suicide' is grammatically incorrect.	updated.
The following sentence does not make sense - line 11-12.	Updated to 'We noted considerable differences in how the aftermath was managed'
Can the authors elaborate on the complicated personal and social factors that influence GPs judgement and decision-making.	Moved this line to the start rather than the end of the paragraph which is describing the factors.
The data under psychological support is interesting but feels a superficial to me. Are there are more data extracts the authors can incorporate to tell a richer story under this theme. Some of the quotations are also short - some of them would benefit from more data in them.	I have added more data extracts and feel it now presents a more in depth and compelling read.
Page 16 line 3 be clear you are talking about a subtheme here.	Updated.
The final theme is clear and coherent and well written. I like the summary.	Thank you.
Discussion I wonder if rumination and personal impact could be combined. The first three themes feel a little superficial and there are interconnections. It may need more analysis or more expansive analytical narrative. A thematic map would help also.	We did wrestle a bit with this one and decided that the rumination theme was so strong that it should be on it's own. I can also accept the position that it is part of the personal reaction. We have decided to accept the suggestion that they are better combined. Rumination has been combined into short term feelings in the personal reaction sub-theme. Including a thematic map greatly helped to organise the themes appropriately and I will be using this tool going forward. it has prompted the changing in order of themes also with barriers to support being placed before psychological support which we feel suits the narrative more appropriately.
'Unperturbability' - should it not be imperturbability.	Corrected.
I think framing the hidden contract as failing with experience and more contact with patients is incorrect. I think it is more that GPs with experience gain more insight into when they need to seek help and form personal support	Updated to reflect these thoughts. We appreciate the insight given.

circles and mechanisms to cope with difficult patients and hard days. GPs still practice with a sense of objectivity because they need to assess and manage patients accurately.	
I would not call one GP account as a resigned fatalism. It could be a deviant extract or anomaly. Would need to be a theme occurring over several samples of GPs to state something that strong about the profession as a whole	This comment is in relation to both sentences which went before in which is was stated that emotional hardening and cynicism was ubiquitous and these transcripts definitely gave this air. I have made it clear that this is but one example of this and also added a quantifying remark to the sentence.
I don't see mention of the strengths and limitations of this study outlined in the discussion	Added.
There are also no future research recommendations mentioned or suggested. Experiences of accessing support would be good as well as GP specific brief interventions for GPs in distress but find where the gaps are.	Conclusions section reworded to make it clear that 'We recommend the provision of a strategy which will facilitate the culture change needed and implement the services needed to support our doctors in general practice, including guidelines for GPs on when and how to ask for help. ' We have also added the recommendation that 'One way to help create that change is to include an opposing view as part of training for all health and social care professionals for example by including in the curriculum modules on how to understand and deal with the effects of self-harm and suicide. '
Conclusion This is written well and I don't have anything to add. I agree with the authors thoughts here. A culture change is needed. In England there is a NHS practitioner health service which I have heard has had a lot of GP self-referrals and allows doctors to seek confidential private help from NHS clinicians in group or one to one format. GPs unfortunately are under immense stress and strain in a wiltering primary care system in the NHS. Workload is rising and resources are reducing. At present, without sustained acute investment, it is not looking bright for the general practice workforce. Unfortunately there have even been a few recent GP suicides which is a tragedy, just like anyone who dies by suicide.	Couldn't agree more.

VERSION 2 – REVIEW

REVIEWER	Sahm, Laura University College Cork, Pharmacy
REVIEW RETURNED	09-Jan-2024

GENERAL COMMENTS	Much improved, minor typos still present e.g. Strengths of this study lie in the qualitative methodology employed in an areas where depth ... however I do not need to see the manuscript again, thank you.
---

REVIEWER	Mughal, Faraz Keele University
REVIEW RETURNED	19-Jan-2024

GENERAL COMMENTS	The authors have done a good job of responding to comments. I could not see the thematic map. You haven't replaced 'identified' with 'generated' in terms of talking about themes as you said you had in the comments. If the above is done, happy to recommend acceptance. An important piece of work.
--

VERSION 2 – AUTHOR RESPONSE

Thank you very much for your response. The minor issues outlined have been addressed including a full proof review for typos and grammar issues. We very much look forward to your response.

many thanks.

Dr Grainne McAnee